# Estrogen Receptor Positive Breast Cancer with High Expression of Androgen Receptor has Less Cytolytic Activity and Worse Response to Neoadjuvant Chemotherapy but Better Survival

**DOI:** 10.3390/ijms20112655

**Published:** 2019-05-30

**Authors:** Maiko Okano, Masanori Oshi, Ali Linsk Butash, Mariko Asaoka, Eriko Katsuta, Xuan Peng, Qianya Qi, Li Yan, Kazuaki Takabe

**Affiliations:** 1Breast Surgery, Department of Surgical Oncology, Roswell Park Comprehensive Cancer Center, Buffalo, NY 14263, USA; tentekomaikocco@hotmail.com (M.O.); masanori.oshi@roswellpark.org (M.O.); ali.butash@roswellpark.org (A.L.B.); mariko.asaoka@roswellpark.org (M.A.); eriko.katsuta@roswellpark.org (E.K.); 2Department of Biostatistics & Bioinformatics, Roswell Park Comprehensive Cancer Center, Buffalo, NY 14263, USA; xuan.peng@roswellpark.org (X.P.); qianya.qi@roswellpark.org (Q.Q.); li.yan@roswellpark.org (L.Y.); 3Department of Surgery, University at Buffalo Jacobs School of Medicine and Biomedical Sciences, The State University of New York, Buffalo, NY 14203-1121, USA; 4Department of Breast Surgery and Oncology, Tokyo Medical University, Tokyo 160-8402, Japan; 5Department of Surgery, Yokohama City University, Yokohama 236-0004, Japan; 6Department of Surgery, Niigata University Graduate School of Medical and Dental Sciences, Niigata 951-8510, Japan; 7Department of Breast Surgery, Fukushima Medical University School of Medicine, Fukushima 960-1295, Japan

**Keywords:** AR, androgen receptor, neoadjuvant chemotherapy, NAC, cytolytic activity, GSEA

## Abstract

Estrogen receptor (ER) positive breast cancer (BC), the most abundant BC subtype, is notorious for poor response to neoadjuvant chemotherapy (NAC). The androgen receptor (AR) was reported to support estradiol-mediated ER activity in an in vitro system. Recently, ER-positive BC with fewer tumor infiltrating lymphocytes (TILs) was shown to have a better prognosis, opposite to the trend seen with ER-negative BC. We hypothesized that ER-positive BC with high expression of AR will have fewer TILs and an inferior response to NAC, but with a better prognosis. In both TCGA and METABRIC cohorts, AR expression was significantly higher in ER-positive BCs compared to ER-negatives (*p* < 0.001, *p* < 0.001, respectively) and it correlated with ER expression (*R* = 0.630, *R* = 0.509, respectively). In ER-positive tumors, AR high tumors enriched UV response down (NES = 2.01, *p* < 0.001), and AR low tumors enriched DNA repair (NES = −2.02, *p* < 0.001). AR high tumors were significantly associated with procancer regulatory T-cells, and AR low tumors were associated with anticancer immune cells, such as CD4, CD8, and Gamma-Delta T-cells and memory B-cells in ER-positive BC (*p* < 0.01). Further, cytolytic activity was significantly lower in AR high BC in both cohorts. Finally, AR high tumors had a significantly lower rate of attaining pathological complete response to NAC (GSE22358), but better survival. In conclusion, our results demonstrated that high AR has fewer tumor infiltrating lymphocytes as well as cytolytic activity and an inferior response to NAC, but better survival in ER-positive BC.

## 1. Introduction

Breast cancer (BC) is the most common cancer among women in the United States excluding skin cancers. About one in eight U.S. women (about 12%) will develop invasive BC in their lifetime [1]. BC is categorized into clinical subtypes by receptor expression statuses with distinctive phenotypes, including varying response to certain treatments [2]. Estrogen receptor (ER) positive BC is the most abundant subtype that accounts for 70% of BC. Although ER-positive BC has a relatively better prognosis compared with the other subtypes [3,4], it is well known to have less tumor infiltrating lymphocytes (TIL) [5,6], and less responsiveness to neoadjuvant chemotherapy (NAC) [7,8]. Interestingly, there have been multiple reports that demonstrated better survival with less TILs in ER-positive BC, which is quite opposite to what is known in ER-negative tumors [9,10].

The androgen receptor (AR) is one of the members of the steroid nuclear receptor family, which includes ER and progesterone receptor (PR). AR is expressed in 50–90% of breast cancers [11,12] and some claim that it is more common than ER or PR [13]. Unlike ER or PR, however, the role of AR in BC progression is mechanistically complex and remains highly controversial. 

AR would rather be famous as the potential target in the therapy for triple negative breast cancer (TNBC), which does not express ER, PR, and HER2/neu and tends to be more aggressive than other subtypes of breast cancers. TNBC is considered not to be a single disease, and is classified into several molecular subtypes, among which there are luminal AR subtypes (LAR) [14]. In vitro experiments have shown that AR is associated with the growth of TNBC cell lines via Wnt/β-catenin signaling and G-protein coupled estrogen receptor activation [15,16]. AR inhibitors are expected to be effective for TNBC patients otherwise lacking molecular targets and several clinical trials have been performed using the AR antagonist for TNBC patients.

On the other hand, several studies have reported the role of AR in ER-positive BC. A study had identified a crosstalk between AR and epidermal growth factor receptor (EGFR), which is involved in cancer cell growth, in ER-positive BC cells [17]. Another report showed that specific domains of AR and ERα physically interact and demonstrated the functional consequences of such interaction using yeast and mammalian two-hybrid systems [18]. AR was also demonstrated to support estradiol-mediated ER activity in BC cells that express both ER and AR [13], which led to clinical trials currently underway using AR-targeting agents (NCT00468715, NCT02007512) [19]. Further, ER-positive BC with high AR expression showed resistance to tamoxifen and aromatase inhibitors in both in vitro and in vivo systems [20,21]. However, several papers that analyzed many patient data concluded that AR is a favorable marker for survival [22,23].

We hypothesized that ER-positive BC with high levels of AR expression attracts TILs and has a worse response to NAC, but better patient survival. In this study, we investigated the associations of AR mRNA expression with ER expression, the number of TILs, the response to NAC, and survival using gene expression data from the publicly available large cohorts.

## 2. Results

### 2.1. Expression of AR mRNA was Significantly Higher in ER-Positive BC

AR expression has been reported to be higher in ER-positive compared with ER-negative BC; however, previous studies used immunohistochemistry (IHC) for AR measurement, which is semiquantitative and the antibodies used were not standardized [12,24,25]. We used transcriptomic data that is rigorously quantitative. AR mRNA expression was significantly lower in basal subtypes among PAM50 classification determined by the gene expression profile (Figure 1A). As expected, ER-positive tumors classified by IHC in TCGA, expressed significantly higher levels of AR mRNA expression when compared to ER-negative patients (Figure 1B). Strikingly, these results were mirrored in the completely independent validation cohort, METABRIC (*p* < 0.001, Figure 1C,D). 

### 2.2. AR mRNA Expression Correlates with ER mRNA Expression

Since several papers have shown the interaction of AR and ER, it was of interest whether high expression of AR mRNA correlates with that of ER mRNA (Figure 2). We found it not only in TCGA (*r* = 0.630, *p* < 0.001), but also in METABRIC (*r* = 0.509, *p* < 0.001) as well.

### 2.3. UV Response Down and DNA Repair Gene Sets Were Enriched in AR High and AR Low Tumors, Respectively, in ER-Positive BC

We conducted GSEA to investigate which gene sets are associated with AR high tumors or AR low tumors in ER-positive BC. High AR expression tumors enriched UV response down (NES = 2.01, *p* < 0.001) hallmark gene sets in GSEA (Figure 3 and Table 1). On the other hand, AR low expression tumors significantly enriched DNA repair (NES = −2.03, *p* < 0.001) hallmark gene sets in GSEA (Figure 3B and Table 2). 

### 2.4. Activated Memory CD4 T Cells and Gamma Delta T cells Were Significantly Lower in AR High Tumors Compared with AR Low Tumors in ER-Positive BC

In order to investigate the relationship between TIL and AR expression in ER-positive BC, immune cell composition was estimated using the CIBERSORT algorithm in TCGA and METABRIC cohorts (Figure 4). We found that AR high tumors associated with significantly low infiltration of CD4 T cells and Gamma Delta T cells (*p* < 0.001) consistently in both TCGA and METABRIC cohorts (Figure 4). The relationship between AR expression and CD8 T cells or B cells were inconsistent in those cohorts. The infiltration of regulatory T cells, which is known to suppress anticancer immunity and negatively impact patient survival, was also inconsistent between the two cohorts. 

### 2.5. Tumor Heterogeneity and Cytolytic Activity Score (CYT) Were Significantly Lower in AR High Compared with AR Low Tumors in ER-Positive BC

The current dogma is that heterogenous tumors generate neoantigens that attract TILs, which are expected to attack cancer with cytolytic activity. Although ER-positive BC is known to have less TILs when compared to triple negative BC, it was of interest to investigate whether this mechanistic model applies in AR high tumors where we found lower infiltration of TILs among ER-positive BC. We found that AR high tumors were significantly associated with low mutant allele tumor heterogeneity (MATH) scores that assess tumor heterogeneity in TCGA whole cohort, but not in any other subtypes, including ER-positive BC (Figure 5 top row). This result implies that infiltration of TILs may not be due to tumor heterogeneity and neoantigen production in AR high tumors among ER-positive BC. On the other hand, CYT was significantly lower in AR high tumors not only in the whole cohort, but also in ER-positive BC and not in the other subtypes, and consistently in both TCGA and METABRIC cohorts (Figure 5 middle and lower rows). This result demonstrates that despite some discrepancy in the infiltration of CD8 T cells and regulatory T cells, AR high tumors possess less overall anticancer immunity. 

### 2.6. AR High Tumors Demonstrated Lower Rates of Attaining Pathological Complete Response (pCR) to Neoadjuvant Chemotherapy (NAC), but Better Survival

In order to explore the relationship between AR expression and response to NAC and subsequent survival, ten NAC cohorts from the GEO database that reported gene expression data and pCR or distant relapse-free survival were analyzed (Figure 6). Although there was only one cohort that reached statistical significance (GSE22358), the pCR rate after NAC was unequivocally lower in AR high tumors in all ten ER-positive BC NAC cohorts (GSE20194, GSE21974, GSE22093, GSE22226, GSE23988, GSE25055, GSE25066, GSE42822, GSE50948). The two ER-positive BC NAC cohorts that have associated survival data, both demonstrated better survival in AR high tumors in ER-positive BC although statistical significance was achieved in only one (GSE25066), which is in agreement with previous reports [26,27]. 

## 3. Discussion

In the present study, we found that AR mRNA expression was higher in ER-positive tumors and the level of AR expression correlated with that of ER expression. AR high tumors enriched UV response down gene sets, whereas AR low tumors enriched DNA repair gene sets. AR high tumors consistently had less infiltration of CD4 and Gamma Delta T cells, but there was a discrepancy with infiltration of CD8 and regulatory T cells between the cohorts. AR high tumors had low tumor heterogeneity only in the whole cohort, whereas CYT was consistently low in the whole cohort and the ER-positive BC cohort in both TCGA and METABRIC. Finally, while AR high tumors demonstrated low pCR rates, they were associated with better survival after NAC. 

Several studies have reported that AR is expressed higher in ER-positive primary BC than ER-negative BC [24,28]. D’Amato et al. demonstrated that estradiol-induced AR-binding sites were enriched for estrogen response elements with significant overlap with ER-binding sites, and that AR is required for maximum ER genomic binding [13]. Our finding that AR expression correlates with ER expression is in agreement with this notion that AR not only interacts with ER but also supports estradiol-mediated ER activity. 

AR was demonstrated to protect prostate cancer cells from DNA damage in vitro [29,30]. AR was also shown to maintain production of DNA damage response molecules in some breast cancer cells and it prevents the accumulation of damaged DNA in vitro [31]. Recent investigations have demonstrated that high AR expression was significantly associated with worse local recurrence-free survival after radiation therapy, which is known to induce DNA damage, and AR can be as a mediator of radioresistance in triple negative breast cancer [32]. Our results are similar to these notions that AR is strongly associated with DNA damage, and AR high tumors associated with chromatin-remodeling genes which were downregulated by UV stress. Our result that AR low tumor significantly enriched DNA repair gene sets aligns with this notion that tumors with AR low expression have more DNA damage because of the lack of the protection by AR and it enriched DNA repair gene sets as a response. This notion is also consistent with our result that MATH score was high in AR low tumor.

Although there are not many articles investigating the relation between TIL and AR, it has been proposed that there is negative correlation between AR expression and immune cell infiltration in patients with HER2 and ER-positive tumors [33]. In the present study, among the infiltrating TILs, CD8 and regulatory T cells, which are known to be the major players, were not consistently low in AR high tumors. Together with the fact that tumor heterogeneity was not significantly lower in AR high tumors in the ER-positive BC cohort, the relationship between AR expression and TIL does not follow the dogma that heterogeneous tumor produces neoantigens that attract TILs [34]. On the other hand, CYT was consistently low in AR high tumors, which was associated with poor response to NAC. To this end, it is speculated that interaction between AR and ER may have some effect on tumor immune microenvironment separate from neoantigen-TIL dogma that suppresses cytolytic activity. 

High infiltration of TILs is a known indicator of a good response to NAC and a better prognosis in the vast majority of cancers, including ER-negative BC [5,6]. This is partially explained by lymphocytes helping chemotherapy to fight cancer cells, and enhancement of anticancer immunity long after chemotherapy is completed. In contrast, recent studies have demonstrated that the opposite trend is the case for the ER-positive BC, where patients’ prognosis after NAC is better with fewer TILs [9,10]. Some argue that the prognosis of breast cancer with high AR is better because these cancers are well differentiated [27]. Our result is strikingly in agreement with these previous reports that AR high patients have better survival despite lower TIL infiltration and worse response to NAC. Based upon our results, we cannot help but speculate that high expression of AR that interacts with ER promotes a mitotic phenotype, less TIL, and less cytolytic activity that leads to NAC resistance. However, enhanced ER activity may have resulted in improvement of response to adjuvant hormonal therapy that consequently prolonged survival.

Our study has some limitations. The first and biggest limitation is that our research has demonstrated only the association between AR expression, TIL, and NAC, but there is no data that indicates causality. The second limitation is that TCGA, METABRIC, and GEO databases have limited numbers of patients and have substantial missing values. In the future, further prospective studies with larger sample volume are needed in order to overcome these issues.

## 4. Materials and Methods

### 4.1. The Cancer Genome Atlas (TCGA) and Molecular Taxonomy of Breast Cancer International Consortium (METABRIC) Patient Cohorts

The gene expression level quantification data (mRNA Z-score of RNA-sequence) of 1093 breast cancer patients, and level three Z-score normalized gene expression data of 1904 patients from METABRIC data were downloaded through cBioPortal as described previously [35,36,37,38]. Gene expression microarray datasets were obtained from ten Gene Expression Omnibus (GEO) (https://www.ncbi.nlm.nih.gov/geo/). Patients were classified as either AR expression high or low group using the median of AR mRNA expression of the individual group. Distant relapse-free survival was determined from time of diagnosis to clinical confirmation of tumor recurrence.

### 4.2. Gene Set Enrichment Analysis (GSEA)

GSEA was calculated by the Broad Institute software (http://software.broadinstitute.org/gsea/index.jsp), as previously described [35,36,37,39]. We classified the patients into two groups according to AR expression using the median of each group.

### 4.3. CIBERSORT, the Immune Cytolytic Activity (CYT) Score, and the Mutant-Allele Tumor Heterogeneity (MATH)

We utilized CIBERSORT, a bioinformatic algorithm, in order to calculate immune cell composition from gene expression profiles, in order to estimate tumor infiltrating immune cells in tumors [40]. Immune cell fraction data was downloaded through The Cancer Imaging Archive (TCIA) (https://tcia.at/home) [41]. The immune cytolytic activity (CYT) score was defined as the geometric mean of granzyme A (GZMA) and perforin (PRF1), which are pivotal cytolytic effectors, with expression values in transcripts per million [42]. The mutant-allele tumor heterogeneity (MATH) level was calculated from the median absolute deviation and the median of its mutant-allele fractions at tumor-specific mutated loci as described previously [43,44]. 

### 4.4. Statistical Analysis

The Kaplan–Meier method with log-rank test was used to determine distant relapse-free survival as previously described [35,36,37,45]. Fisher’s exact test was used to determine if there were nonrandom associations between two categorical variables. Student’s *t*-test was used to analyze the differences between continuous values. In all analyses, a two-sided *p* < 0.05 was considered statistically significant. All statistical analyses were performed using Microsoft Excel 2010, R software (http:///www.rproject.org/), and Bioconductor (http://bioconductor.org/).

## 5. Conclusions

Our results demonstrated that high AR has fewer tumor infiltrating lymphocytes and a worse response to NAC, but better survival in ER-positive BC. The fact that AR expression plays some cancer-immunological role in ER-positive breast cancer is enhanced by this report, which can add a novel aspect to future research and treatment of ER-positive breast cancer.

## Figures and Tables

**Figure 1 ijms-20-02655-f001:**
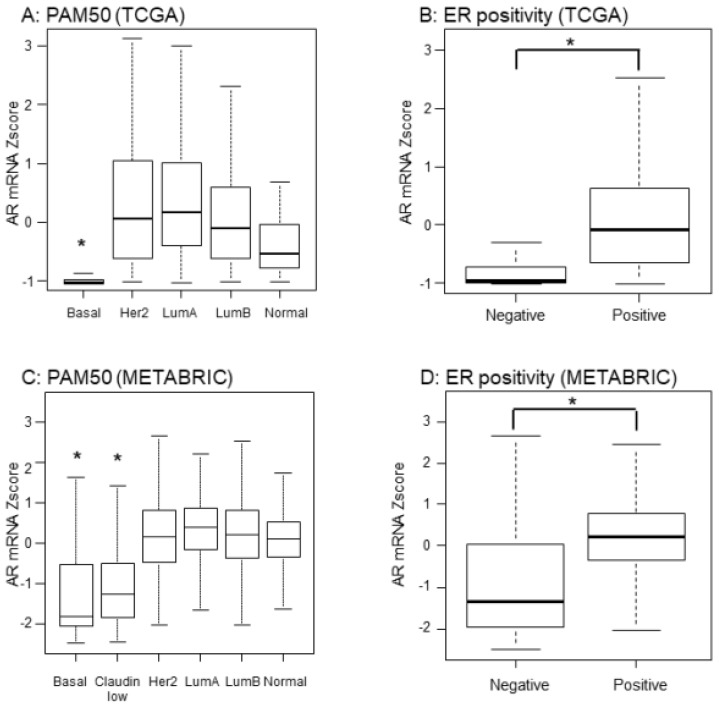
Expression levels of AR mRNA in TCGA and METABRIC cohorts. AR mRNA levels were quantified by mRNA Z-scores of RNA-sequences. Bold lines demonstrate the median and box plot is 95% interval. * *p* < 0.001. (**A**) PAM50 subtype in TCGA; (**B**) ER positivity defined by IHC in TCGA; (**C**) PAM50 subtype in METABRIC; (**D**) ER positivity defined by IHC in METABRIC.

**Figure 2 ijms-20-02655-f002:**
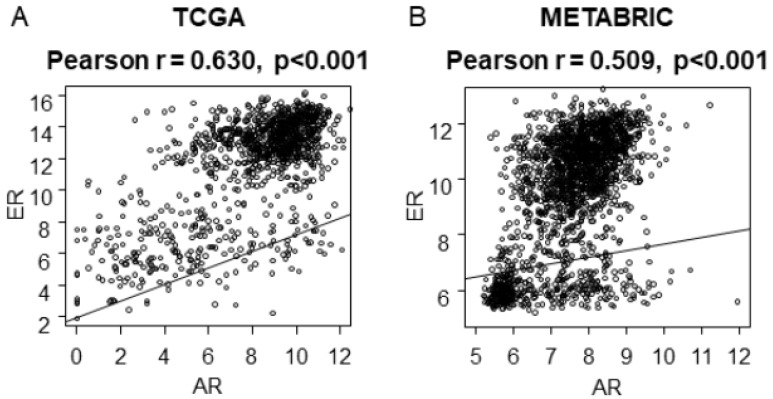
Pearson correlation coefficient (r) and *p*-value (p) between AR mRNA expression and ER mRNA expression. (**A**) A scatter plot of TCGA cohort; (**B**) A scatter plot of METABRIC cohort.

**Figure 3 ijms-20-02655-f003:**
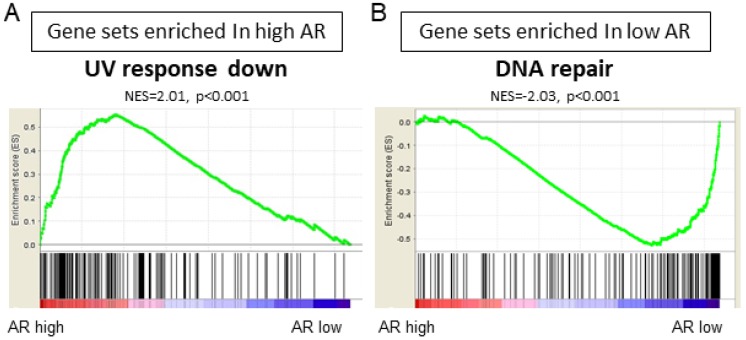
Gene set enrichment analysis (GSEA) demonstrated significant enrichment of the following hallmark gene sets to AR mRNA expression in ER-positive BC in TCGA. (**A**) UV response down gene set enriched in high expression of AR mRNA; (**B**) DNA repair gene set enriched in low expression of AR mRNA.

**Figure 4 ijms-20-02655-f004:**
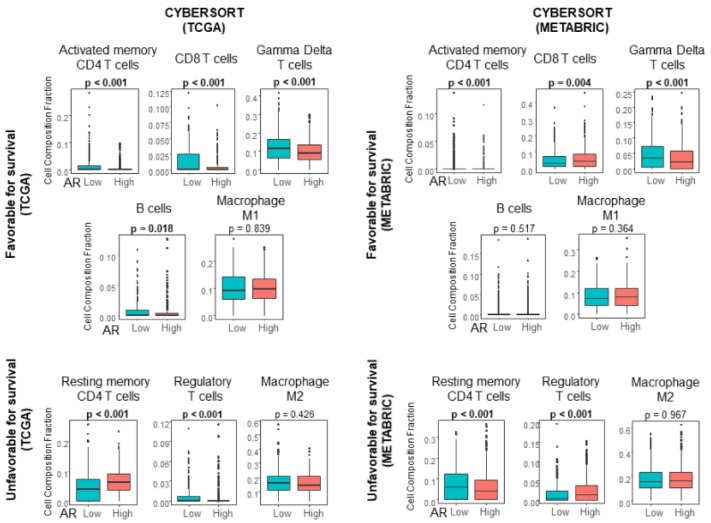
In CIBERSORT analysis, activated memory CD4 T cells and Gamma Delta T cells in AR high tumors are lower than those in AR low tumors in ER-positive BC.

**Figure 5 ijms-20-02655-f005:**
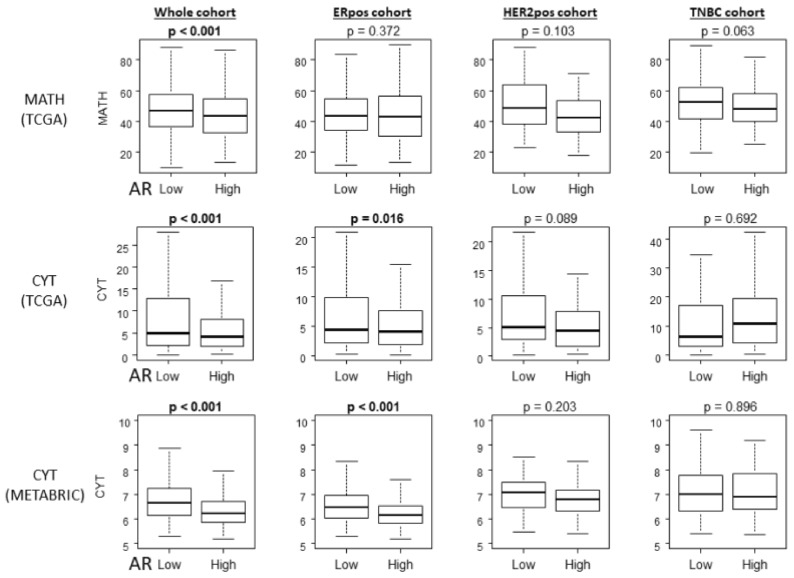
Tumor heterogeneity and cytolytic activity score in AR high BC is lower than that in AR low ER-positive BC.

**Figure 6 ijms-20-02655-f006:**
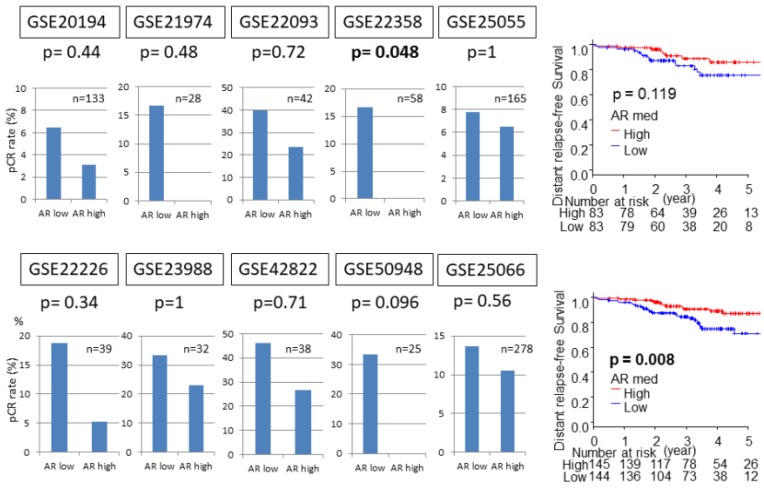
pCR rate and Kaplan–Meier survival curve by AR mRNA levels of GEO cohorts. High and low expressions are represented by the red and blue lines, respectively.

**Table 1 ijms-20-02655-t001:** Significant gene sets that associated with high expression of AR using Gene Set Enrichment Analysis (GSEA)(TCGA).

Name	Size	ES	NES	NOM *p*-val	FDR *q*-val	FWER *p*-val
HALLMARK_UV_RESPONSE_DN	140	0.5521	2.0113	0.0000	0.0126	0.0210

1 ES, enrichment score; 2 NES, normalized enrichment score; 3 NOM-*p*-val, normalized *p*-value; 4 FDR *q*-val, false discovery rate *q*-value; 5 FWER *p*-val, family-wise error rate *p*-value.

**Table 2 ijms-20-02655-t002:** Significant gene sets that associated with low expression of AR using Gene Set Enrichment Analysis (GSEA)(TCGA).

Name	Size	ES	NES	NOM *p*-val	FDR *q*-val	FWER *p*-val
HALLMARK_DNA_REPAIR	142	−0.5265	−2.0298	0.0000	0.0281	0.0150

1 ES, enrichment score; 2 NES, normalized enrichment score; 3 NOM-*p*-val, normalized *p*-value; 4 FDR *q*-val, false discovery rate *q*-value; 5 FWER *p*-val, family-wise error rate *p*-value.

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
