# Peer review of "Estrogen Receptor Positive Breast Cancer with High Expression of Androgen Receptor has Less Cytolytic Activity and Worse Response to Neoadjuvant Chemotherapy but Better Survival"

_ijms, 2019, doi:10.3390/ijms20112655_

Round 1

Reviewer 1 Report

Authors should refer to results obtained in vivo and in vitro experiments and emphasize the correlation between their results and the literature.

Introduction and conclusion sections are scantly. Authors  can improve it by reading a recently published review on AR action in breast cancers. doi: 10.3389/fendo.2018.00492

Pragraphs 2.1 and 2.2 can be summarized in a paragraph avoiding redundant information.

Author Response

Point 1: Authors should refer to results obtained in vivo and in vitro experiments and emphasize the correlation between their results and the literature.

Response 1: We really appreciate Reviewer 1 for giving us the suggestion. We have revised our discussion as below.

Several studies have reported that AR is expressed higher in ER positive primary BC than ER negative BC (24, 28). D'Amato et al demonstrated that estradiol-induced AR-binding sites were enriched for estrogen response elements with significant overlap with ER-binding sites, and that AR is required for maximum ER genomic binding (13). Our finding that AR expression correlates with ER expression is in agreement with this notion that AR not only interacts with ER but also supports estradiol-mediated ER activity.

AR was demonstrated to protect prostate cancer cells from DNA damage in vitro (29, 30). AR was also shown to maintain production of DNA damage response molecules in some breast cancer cells and it prevents the accumulation of damaged DNA in vitro (31). Recent investigations have demonstrated that high AR expression was significantly associated with worse local recurrence-free survival after radiation therapy, which known to induce DNA damage, and AR can be as a mediator of radioresistance although in triple negative breast cancer (32). Our results are similar to these notions that AR is strongly associated with DNA damage, and AR high tumors associated with chromatin-remodeling genes which were down-regulated by UV stress. Our result that AR low tumor significantly enriched DNA repair gene sets aligns with this notion that tumors with AR low expression have more DNA damage because of the lack of the protection by AR and it enriched DNA repair gene sets as a response. This notion is also consistent with our result that MATH score was high in AR low tumor.

Although there are not many articles that investigated the relation between TIL and AR, it has been proposed that there is negative correlation between AR expression and immune cell infiltration in patients with HER2 and ER positive tumors (33). This is in agreement with our result that CYT, the cytolytic activity of immune cells, was consistently low in AR high tumors. To this end, it is speculated that interaction between AR and ER may suppress tumor immune microenvironment that resulted in poor response to NAC.

High infiltration of TILs is a known indicator of a good response to NAC and a better prognosis in the vast majority of cancers, including ER negative BC (5, 6). This is partially explained by lymphocytes helping chemotherapy to fight cancer cells, and enhancement of anti-cancer immunity long after chemotherapy is completed. In contrast, recent studies have demonstrated that the opposite trend is the case for the ER positive BC, where patients’ prognosis after NAC is better with fewer TILs (9, 10). Some argue that the prognosis of breast cancer with high AR is better because these cancers are well differentiated (27). Our result is strikingly in agreement with these previous reports that AR high patients have better survival despite lower TIL infiltration and worse response to NAC. Based upon our results, we cannot help but speculate that high expression of AR that interacts with ER promotes a mitotic phenotype, less TIL and less cytolytic activity that leads to NAC resistance. However, enhanced ER activity may have resulted in improvement of response to adjuvant hormonal therapy that consequently prolonged survival.

Point 2: Introduction and conclusion sections are scantly. Authors can improve it by reading a recently published review on AR action in breast cancers. doi: 10.3389/fendo.2018.00492

Response 2: Thank you for this important suggestion. We added the information from the review as below.

Breast cancer (BC) is the most common cancer among women in the United States excluding skin cancers. About 1 in 8 U.S. women (about 12%) will develop invasive BC in their lifetime (1). BC is categorized into clinical subtypes by receptor expression statuses with distinctive phenotypes, including varying response to certain treatments (2). Estrogen receptor (ER) positive BC is the most abundant subtype that accounts for 70% of BC. Although ER positive BC has a relatively better prognosis compared with the other subtypes (3, 4), it is well known to have less tumor infiltrating lymphocytes (TIL) (5, 6), and less responsiveness to neoadjuvant chemotherapy (NAC) (7, 8). Interestingly, there have been multiple reports that demonstrated better survival with less TILs in ER positive BC, which is quite opposite to what is known in ER negative tumors (9, 10).

The androgen receptor (AR) is one of the members of the steroid nuclear receptor family, which includes ER and progesterone receptor (PR). AR is expressed in 50–90% of breast cancers (11, 12) and some claim that it is more common than ER or PR (13). Unlike ER or PR, however, the role of AR in BC progression is mechanistically complex and remains highly controversial.

AR has been proposed as the potential therapeutic target for triple negative breast cancer (TNBC), which does not express ER, PR and HER2/neu and tends to be more aggressive than other subtype of breast cancers. TNBC is considered not to be a single disease, and is classified into several molecular subtypes, among which there is luminal AR subtypes (LAR) (14). In vitro experiments have shown that AR is associated with the growth of TNBC cell lines via Wnt/β-catenin signaling and G-protein coupled estrogen receptor activation (15, 16). Several clinical trials have been performed using the AR antagonist for TNBC patients.

On the other hand, several studies have reported the role of AR in ER positive BC. A study had identified a crosstalk between AR and epidermal growth factor receptor (EGFR), which is involved in cancer cell growth, in ER positive BC cells (17). Other report had shown that specific domains of AR and ERα physically interact and have demonstrated the functional consequences of such interaction using the yeast and mammalian two-hybrid systems (18). AR was also demonstrated to support estradiol-mediated ER activity in BC cells that express both ER and AR (13), which led to clinical trials currently underway using AR-targeting agents (NCT00468715, NCT02007512) (19). Further, ER positive BC with high AR expression showed resistance to tamoxifen and aromatase inhibitors in both in vitro and in vivo systems (20, 21). However, several papers that analyzed many patient data concluded that AR is a favorable marker for survival (22, 23).

We hypothesized that ER positive BC with high levels of AR expression attracts TILs and has a worse response to NAC, but better patient survival. In this study, we investigated the associations of AR mRNA expression with ER expression, the amount of TILs, the response to NAC, and survival using gene expression data from the publicly available large cohorts.

Point 3: Paragraphs 2.1 and 2.2 can be summarized in a paragraph avoiding redundant information.

Response 3: We appreciate this comment. We tried to shorten this section as below.

2.1. Expression of AR mRNA was significantly higher in ER positive BC.

AR expression has been reported to be higher in ER positive compared with ER negative BC, however, previous studies used immunohistochemistry (IHC) for AR measurement that is semi-quantitative and the antibodies used were not standardized (12, 24, 25). We used transcriptomic data that is rigorously quantitative. AR mRNA expression was significantly lower in basal subtypes among PAM50 classification determined by the gene expression profile (Figure 1A). As expected, ER positive tumors classified by IHC in TCGA, expressed significantly higher levels of AR mRNA expression when compared to ER negative patients (Figure 1B). Strikingly, these results were mirrored in the completely independent validation cohort, METABRIC (p<0.001, Figure 1C, 1D).

2.2. AR mRNA expression correlates with ER mRNA expression

Since several papers have shown the interaction of AR and ER, it was of interest whether high expression of AR mRNA correlates with that of ER mRNA (Figure 2). We found it not only in TCGA (r=0.630, p<0.001), but also in METABRIC (r=0.509, p<0.001) as well.

Reviewer 2 Report

It's a well characterized and well-written manuscript and should be accepted as it is.

Author Response

Point 1: It's a well characterized and well-written manuscript and should be accepted as it is.

Response 1: We really appreciate Reviewer 2 for taking his/her time.

Reviewer 3 Report

Introduction and conclusion section must be improved. In particular, deepen the description of triple negative (ER-PR-EGFR-) breast cancers and the relation with AR is appreciate. To this end, authors can read recently published reviews focusing on the role of AR in breast Cancer.

Additionally, conclusion section should be improved highlighting the impact of results obtained in therapy and etc.

Author Response

Point 1: Introduction and conclusion section must be improved. In particular, deepen the description of triple negative (ER-PR-EGFR-) breast cancers and the relation with AR is appreciate. To this end, authors can read recently published reviews focusing on the role of AR in breast Cancer.

Response 1: We appreciate Reviewer 3 for indicating an important view. We have revised our manuscript accordingly.

Breast cancer (BC) is the most common cancer among women in the United States excluding skin cancers. About 1 in 8 U.S. women (about 12%) will develop invasive BC in their lifetime (1). BC is categorized into clinical subtypes by receptor expression statuses with distinctive phenotypes, including varying response to certain treatments (2). Estrogen receptor (ER) positive BC is the most abundant subtype that accounts for 70% of BC. Although ER positive BC has a relatively better prognosis compared with the other subtypes (3, 4), it is well known to have less tumor infiltrating lymphocytes (TIL) (5, 6), and less responsiveness to neoadjuvant chemotherapy (NAC) (7, 8). Interestingly, there have been multiple reports that demonstrated better survival with less TILs in ER positive BC, which is quite opposite to what is known in ER negative tumors (9, 10).

The androgen receptor (AR) is one of the members of the steroid nuclear receptor family, which includes ER and progesterone receptor (PR). AR is expressed in 50–90% of breast cancers (11, 12) and some claim that it is more common than ER or PR (13). Unlike ER or PR, however, the role of AR in BC progression is mechanistically complex and remains highly controversial.

AR has been proposed as the potential therapeutic target for triple negative breast cancer (TNBC), which does not express ER, PR and HER2/neu and tends to be more aggressive than other subtype of breast cancers. TNBC is considered not to be a single disease, and is classified into several molecular subtypes, among which there is luminal AR subtypes (LAR) (14). In vitro experiments have shown that AR is associated with the growth of TNBC cell lines via Wnt/β-catenin signaling and G-protein coupled estrogen receptor activation (15, 16). Several clinical trials have been performed using the AR antagonist for TNBC patients.

On the other hand, several studies have reported the role of AR in ER positive BC. A study had identified a crosstalk between AR and epidermal growth factor receptor (EGFR), which is involved in cancer cell growth, in ER positive BC cells (17). Other report had shown that specific domains of AR and ERα physically interact and have demonstrated the functional consequences of such interaction using the yeast and mammalian two-hybrid systems (18). AR was also demonstrated to support estradiol-mediated ER activity in BC cells that express both ER and AR (13), which led to clinical trials currently underway using AR-targeting agents (NCT00468715, NCT02007512) (19). Further, ER positive BC with high AR expression showed resistance to tamoxifen and aromatase inhibitors in both in vitro and in vivo systems (20, 21). However, several papers that analyzed many patient data concluded that AR is a favorable marker for survival (22, 23).

We hypothesized that ER positive BC with high levels of AR expression attracts TILs and has a worse response to NAC, but better patient survival. In this study, we investigated the associations of AR mRNA expression with ER expression, the amount of TILs, the response to NAC, and survival using gene expression data from the publicly available large cohorts.

Point 2: Additionally, conclusion section should be improved highlighting the impact of results obtained in therapy and etc.

Response 2: Thank you for this important suggestion. We have revised the section as below.

Our results demonstrated that high AR has fewer tumor infiltrating lymphocytes and a worse response to NAC, but better survival in ER positive BC. The fact that AR expression plays some cancer-immunological role in ER positive breast cancer is enhanced by this report, which can add a novel aspect to future research and treatment of ER positive breast cancer.